# GPU-Accelerated Primal Learning for Extremely Fast Large-Scale Classification

**John T. Halloran**
Department of Public Health Sciences
University of California, Davis
jthalloran@ucdavis.edu

**David M. Rocke**
Department of Public Health Sciences
University of California, Davis
dmrocke@ucdavis.edu

## Abstract

One of the most efficient methods to solve $L_2$-regularized primal problems, such as logistic regression and linear support vector machine (SVM) classification, is the widely used trust region Newton algorithm, *TRON* [39]. While *TRON* has recently been shown to enjoy substantial speedups on shared-memory multi-core systems [36, 22], exploiting graphical processing units (GPUs) to speed up the method is significantly more difficult, owing to the highly complex and heavily sequential nature of the algorithm. In this work, we show that using judicious GPU-optimization principles, *TRON* training time for different losses and feature representations may be drastically reduced. For sparse feature sets, we show that using GPUs to train logistic regression classifiers in LIBLINEAR is up to an order-of-magnitude faster than solely using multithreading. For dense feature sets–which impose far more stringent memory constraints–we show that GPUs substantially reduce the lengthy SVM learning times required for state-of-the-art proteomics analysis, leading to dramatic improvements over recently proposed speedups. Furthermore, we show how GPU speedups may be mixed with multithreading to enable such speedups when the dataset is too large for GPU memory requirements; on a massive dense proteomics dataset of nearly a quarter-billion data instances, these mixed-architecture speedups reduce SVM analysis time from over half a week to less than a single day while using limited GPU memory.

## 1 Introduction

Over the past decade, GPUs have become valuable computing resources to accelerate the training of popular machine learning models, playing a key role in the widespread use of deep models and the growing ecosystem of deep learning packages [9, 11, 1, 30, 44]. When a training algorithm admits an efficient GPU implementation (such as gradient boosted trees [42], nonlinear kernel learning [8, 50], and primal methods like *L-BFGS* [40] and variants of gradient descent), the speedups gained using GPUs, as opposed to only CPUs, are often substantial. For instance, in PyTorch [44], training a logistic regression classifier on the rcv1 [37] dataset with gradient descent is 14.6 times faster using a Tesla V100 GPU versus using 24 CPU threads with an Intel Xeon Gold 5118 (similarly, training with *L-BFGS* in this example is 13.1 times faster using the V100, detailed in [23]).

Specialized solvers commonly provide even more speed. For the previous logistic regression example, using just a single CPU thread with scikit-learn's [45] *TRON* solver–the primal learning algorithm for logistic regression and SVM classification/regression adapted from LIBLINEAR [13]–is 94.7 and 10.9 times faster than GPU-accelerated gradient descent and *L-BFGS*, respectively, implemented in PyTorch. However, while significant work has been done to further accelerate *TRON* and many other extremely fast machine learning solvers [27, 54, 35, 31, 32] using multiple CPU cores [6, 29, 28, 36, 10, 46, 55, 22], analogous GPU speedups for such efficient algorithms are often lacking.

This lack of GPU exploitation is due to the specialized structure and complexity of these algorithms, which **naturally lend themselves to multithreaded speedups** on shared memory systems, **yet resist optimizations on GPU architectures**.

For example, *TRON* relies on random access to features for SVM losses, which is naturally supported in multithreaded systems, but prevents memory coalescing (and is thus deleterious) for GPU computation. Furthermore, large memory transfers between the CPU and GPU are expensive, so that the complex, sequential dependency of variables in specialized algorithms like *TRON* make optimal GPU use difficult. Indeed, we show that while most of the computational bottlenecks for logistic regression in *TRON* are linear algebra operations [36] (for which GPUs greatly outperform CPUs), using *TRON*'s original variable access pattern in LIBLINEAR results in poor GPU performance–performing even worse than using only a single CPU thread on one of the presented datasets.

Herein, we show that using just a single GPU, excellent training speedups are achievable by overly CPU-specialized machine learning algorithms such as *TRON*. In particular, for different feature representations and loss functions, we show that *TRON* training times may be drastically reduced using judicious GPU-optimization principles.

**Sparse Features.** For sparse feature representations, we successively optimize *TRON* for logistic regression (referred to as *TRON*-LR) in LIBLINEAR using several strategies to: a) decouple the sequential dependence of variables, b) minimize the number of large-memory transfers between GPU and CPU, and c) maximize parallelism between the CPU and GPU. We show that while *TRON*'s original variable access pattern limits the effectiveness of GPU computation, using a single CPU thread with our GPU optimizations results in a 70.8% improvement in training time (averaged over the presented datasets) over the single-threaded version of *TRON* in standard LIBLINEAR. In addition, we show that mixing our GPU optimizations with multithreading provides further speedups, resulting in an average **89.2% improvement over single-thread optimized *TRON*** and an average **65.2% improvement over *TRON* in the multithread-optimized version of** LIBLINEAR [36].

**Dense Features.** For dense feature representations, we show that GPUs substantially reduce SVM learning times for state-of-the-art analysis of dense proteomics datasets [33]. Overcoming the random access restrictions of *TRON* SVM learning (referred to as *TRON*-SVM), we show that using just a single GPU leads to an average as much as triples the performance of recently proposed speedups for this application [22]. **On a large-scale dataset of over 23 million data instances, these GPU speedups reduce SVM learning time from 14.4 hours down to just 1.9 hours**. Furthermore, dense feature sets impose stringent GPU memory constraints, particularly for the massive datasets regularly produced in biological experiments. Thus, we demonstrate how GPU optimizations may be mixed with multithreading to significantly reduce GPU memory constraints. On a massive proteomics dataset consisting of over 215 million data instances–which exceeds memory requirements for GPU-only speedups–these mixed-architecture speedups drastically outperform recent multithread-optimized solvers, **reducing standard analysis time from 4.4 days down to just 19.7 hours**.

The paper is organized as follows. In Section 2, we describe relevant previous work speeding up *TRON* for both sparse and dense feature sets on shared memory systems. In Section 3, we define the general *TRON* algorithm and computational bottlenecks encountered minimizing different loss functions. In Sections 4 and 5, we discuss how the computational bottlenecks in algorithms like *TRON* natively resist GPU speedups, and GPU-optimization principles to overcome these hurdles (providing the resulting GPU optimizations for the objectives and feature architectures under study). We demonstrate that the presented GPU-optimizations drastically outperform recent multithreaded speedups in Section 6, and conclude with future avenues extending the presented work to other high-performance GPU packages (such as PyTorch) in Section 7.

## 2 Previous Work

Serving as the primal solver in the popular package LIBLINEAR [13], *TRON* has been extensively tested and shown to enjoy superior speed and convergence compared to other second-order solvers, such as the widely-used quasi-Newton algorithm *L-BFGS* [40] and the modified Newton algorithm *L2-SVM-MFN* [35] (one of the fastest algorithms for large-scale primal SVM learning). As a Newton method, the algorithm enjoys general quadratic convergence without loading the entire Hessian into memory, thus only using linear memory. For logistic and SVM losses in LIBLINEAR, *TRON*'s convergence speed has further been theoretically improved by refining trust-region update rules [26] and applying a preconditioner matrix to help stabilize optimization [25]. In [36], multithreaded

optimizations in shared-memory multi-core systems were extensively explored to speed up *TRON*'s computational bottlenecks (further described in Section 3) for logistic regression. Evaluating several multithreading libraries (i.e., OpenMP, Intel's Math Kernel Library, and the sparse matrix multiplication package librsb) over a large number of datasets, OpenMP was found to provide the best multithreaded performance and was subsequently integrated into the multi-core release of LIBLINEAR.

## 2.1 SVM Classification Using *TRON* for Fast Large-Scale Proteomics Analysis

In proteomic analysis pipelines, SVM classification using Percolator [33] is a critical step towards accurately analyzing protein data collected via tandem mass spectrometry (MS/MS). Given a collection of MS/MS spectra representing the protein subsequences (called *peptides*) present in a biological sample, the first stage of proteomics analysis typically consists of identifying the input spectra by searching (i.e., scoring and ranking) a database of peptides. This first stage thus results in a list of *peptide-spectrum-matches* (*PSMs*) and their respective scores. In practice, however, database-search scoring functions are often poorly calibrated, making PSMs from different spectra difficult to compare and diminishing overall identification accuracy. To correct for this, the list of PSMs, as well as dense feature vectors describing each match, are fed into Percolator for *recalibration*.

Percolator first estimates PSM labels (i.e., correct versus incorrect) using false discovery rate analysis [34], then trains a linear SVM to classify correct and incorrect identifications. These two steps are iterated until convergence and the input PSM scores are subsequently recalibrated using the final learned SVM parameters. Furthermore, to prevent overfitting and improve generalizability within each iteration, three-fold cross-validation is carried out over three disjoint partitions of the original dataset, followed by further nested cross-validation within each fold [16].

The accuracy improvements of Percolator recalibration have been well demonstrated for a wide variety of PSM scoring functions–e.g., linear [33, 7, 52], *p*-value based [15, 24, 38], and dynamic Bayesian networks [18, 19, 17]–and complex PSM feature sets–e.g., Fisher kernels [20, 21], subscores of linear functions [47], ensembles of scoring functions [49], and features derived using deep models [14]. However, due to the iterative training of many SVMs during cross-validation, Percolator requires substantial analysis times for large-scale datasets commonly collected in MS/MS experiments. Initial work sought to speed up Percolator runtimes by randomly sampling a small portion of the data to train over [41], but this was subsequently shown to unpredictably diminish the performance of learned parameters [22]. Thus, to combat these lengthy analysis times without affecting learned SVM parameters, recent work [22] applied extensive systems-level speedups and multithreading in both Percolator's original primal solver, *L2-SVM-MFN*, and *TRON* (heavily optimized to utilize dense feature vectors). While both optimized solvers were shown to significantly improve Percolator training times for large-scale data, *TRON* displayed markedly superior performance.

## 3   Trust Region Newton Methods for Primal Classification

Consider feature vectors $\boldsymbol{x}_i \in \mathbb{R}^n, i = 1, \ldots, l$ and label vector $\boldsymbol{y} \in \{-1, 1\}^l$, and let $X = [\boldsymbol{x}_1 \ldots \boldsymbol{x}_l]^T$ be the feature matrix. For vectors, index-set subscripts denote subvectors and for matrices, pairs of index-set subscripts denote submatrices. The general objective, which we wish to minimize w.r.t. $\boldsymbol{w}$, is

$$f(\boldsymbol{w}) = \frac{1}{2}\|\boldsymbol{w}\|_2^2 + C \sum_{i=1}^{l} \ell(\boldsymbol{w}; \boldsymbol{x}_i, y_i), \tag{1}$$

where $\frac{1}{2}\|\boldsymbol{w}\|_2^2$ is the regularization term, $C > 0$ is a regularization hyperparameter, and $\ell(\boldsymbol{w}; \boldsymbol{x}_i, y_i)$ is a loss function.

When $\ell(\boldsymbol{w}; \boldsymbol{x}_i, y_i) = \log(1 + \exp(-y_i \boldsymbol{w}^T \boldsymbol{x}_i))$, commonly referred to as the *logistic loss*, minimizing Equation 1 corresponds to learning a classifier using logistic regression. Similarly, minimizing Equation 1 when $\ell(\boldsymbol{w}; \boldsymbol{x}_i, y_i) = (\max(0, 1 - y_i \boldsymbol{w}^T \boldsymbol{x}_i))^2$, commonly referred to as the *quadratic SVM* or *L2-SVM loss*, corresponds to learning a linear SVM classifier. We denote Equation 1 under the logistic loss as $f_{\mathrm{LR}}(\boldsymbol{w})$ and, under the L2-SVM loss, as $f_{\mathrm{L2}}(\boldsymbol{w})$.

---

**Algorithm 1** The *TRON* algorithm

1: Given $w$, $\Delta$, and $\sigma_0$
2: Calculate $f(\boldsymbol{w})$                         *// Critically depends on $\boldsymbol{z} = X^T \boldsymbol{w}$*
3: **while** Not converged **do**
4:      Find $\boldsymbol{d} = \arg\min_{\boldsymbol{v}} q(\boldsymbol{v})$ s.t. $\|\boldsymbol{v}\|_2 \leq \Delta$.        *// Critically depends on $\nabla f(\boldsymbol{w})$, $\nabla^2 f(\boldsymbol{w})\boldsymbol{v}$*
5:      Calculate $f(\boldsymbol{w}+\boldsymbol{d}), \sigma = \frac{f(\boldsymbol{w}+\boldsymbol{d})-f(\boldsymbol{w})}{q(\boldsymbol{d})}$    *// Critically depends on $\boldsymbol{z} = X^T(\boldsymbol{w}+\boldsymbol{d})$*
6:      **if** $\sigma > \sigma_0$ **then**
7:          $\boldsymbol{w} \leftarrow \boldsymbol{w} + \boldsymbol{d}$, increase trust region $\Delta$.
8:      **else**
9:          Shrink $\Delta$.
10:      **end if**
11: **end while**

---

*TRON* is detailed in Algorithm 1. At each iteration, given the current parameters $\boldsymbol{w}$ and trust region interval $\Delta$, TRON considers the following quadratic approximation between function parameters,

$$f(\boldsymbol{w} + \boldsymbol{d}) - f(\boldsymbol{w}) \approx q(\boldsymbol{d}) \equiv \nabla f(\boldsymbol{w})^T \boldsymbol{d} + \frac{1}{2}\boldsymbol{d}^T \nabla^2 f(\boldsymbol{w})\boldsymbol{d}. \tag{2}$$

A truncated Newton step ($\boldsymbol{d}$ on line 4 in Algorithm 1), confined in the trust region, is then found using a conjugate gradient procedure. If $q(\boldsymbol{d})$ is close to $f(\boldsymbol{w}+\boldsymbol{d}) - f(\boldsymbol{w})$, $\boldsymbol{w}$ is updated to $\boldsymbol{w}+\boldsymbol{d}$ and the trust region interval is increased for the subsequent iteration. Otherwise, $\boldsymbol{w}$ remains unchanged and the trust region interval is shrunk.

Note that the function evaluation $f(\boldsymbol{w})$–which critically depends on computing $\boldsymbol{z} = X^T w$ for both losses– must be computed for each new iteration, as well as the gradient and Hessian for Equation 2. However, computing only the Hessian-vector product in Equation 2 avoids loading the entire Hessian into memory (which would be intractable for large datasets). Thus, the most intensive portions of *TRON* are the computations of $\boldsymbol{z} = X^T w, \nabla f(\boldsymbol{w})$, and $\nabla^2 f(\boldsymbol{w})\boldsymbol{v}$ (where $\boldsymbol{v}$ is the optimization variable in line 4 of Algorithm 1), summarized for both losses in Table 1. Further derivation of these quantities is available in [23].

We note that arbitrary loss functions (and combinations thereof) may be used in Equation 1, thus allowing future work utilizing the highly efficient *TRON* in popular automatic differentiation [5] packages [44, 1, 48, 43]. However, these packages rely on GPUs for optimal performance, the use of which *TRON* natively resists (as we'll see, and rectify, for the two loss functions considered).

| **Logistic Loss** | **L2-SVM Loss** |
|---|---|
| $\boldsymbol{z} = X^T w$, to compute $f_{\text{LR}}(\boldsymbol{w})$ | $\boldsymbol{z} = X^T w$, to compute $f_{\text{L2}}(\boldsymbol{w})$ |
| $\nabla f_{\text{LR}}(\boldsymbol{w}) = \boldsymbol{w} + C\sum_{i=1}^{l}(h(y_i\boldsymbol{z}_i) - 1)y_i\boldsymbol{x}_i$, where $h(y_i\boldsymbol{z}_i) = (1 + e^{-y_i\boldsymbol{z}_i})^{-1}$ | $\nabla f_{\text{L2}}(\boldsymbol{w}) = \boldsymbol{w} + 2CX_{I,:}^T(\boldsymbol{z}_I - \boldsymbol{y}_I)$, where $I \equiv \{i\|1 - y_i\boldsymbol{z}_i > 0\}$ is an index set and the operator : denotes all elements along the corresponding dimension (i.e., all columns in this case) |
| $\nabla^2 f_{\text{LR}}(\boldsymbol{w})\boldsymbol{v} = \boldsymbol{v} + CX^T(D(X\boldsymbol{v}))$, where $D$ is a diagonal matrix with elements $D_{i,i} = h(y_i\boldsymbol{z}_i)(1 - h(y_i\boldsymbol{z}_i))$ | $\nabla^2 f_{\text{L2}}(\boldsymbol{w})\boldsymbol{v} = \boldsymbol{v} + 2CX_{I,:}^T(X_{I,:}\boldsymbol{v})$ |

Table 1: *TRON* major bottleneck computations for logistic and L2-SVM losses.

## 4   Accelerating *TRON*-LR training using GPUs

Assume a shared-memory multi-core system and a single GPU with sufficient memory for the variables in Table 1 (this is later relaxed in Section 5). Herein, the CPU is referred to as the *host* and the GPU is referred to as the *device*.

*TRON*-LR runtime is dominated by three major matrix-vector multiplications in the bottleneck computations listed in Table 1: $\boldsymbol{z} = X^T w, \nabla^2 f_{\text{LR}}(\boldsymbol{w})\boldsymbol{v}$, and $\nabla f_{\text{LR}}(\boldsymbol{w}) = \boldsymbol{w} + CX\hat{\boldsymbol{z}}$, where $\hat{\boldsymbol{z}}_i = (h(y_i\boldsymbol{z}_i) - 1)y_i$. For instance, profiling *TRON*-LR in LIBLINEAR training on the large-scale

`SUSY` [4] dataset, these **three matrix-vector multiplications account for 82.3% of total training time**. We thus first attempt to accelerate *TRON*-LR by computing these quantities quickly on the device (as was similarly done in [36] using multithreading).

In LIBLINEAR, this first attempt at GPU acceleration (called *TRON*-LR-GPU$^0$) is implemented using cuSPARSE to perform sparse linear algebra operations as efficiently as possible for LIBLINEAR's sparse feature representation. Compared to the standard single-threaded implementation of LIBLINEAR on the `SUSY` dataset, *TRON*-LR-GPU$^0$ achieves a speedup of 0.65–*TRON*-LR-GPU$^0$ is actually slower than the single-threaded LIBLINEAR! *TRON*-LR-GPU$^0$ fairs better on other presented datasets, but performs poorly overall (displayed in Figure 1).

## 4.1 Sequentially Dependent Variables

The critical issue encountered by *TRON*-LR-GPU$^0$ is *TRON*'s overly sequential dependency of variables; once variable vectors are computed on the GPU, they are immediately needed on the host CPU to proceed with the next step of the algorithm. For instance, computing the bottleneck $z = X^T(\boldsymbol{w} + \boldsymbol{d})$ using cuSPARSE is fast, but $z$ must immediately be transferred back to the host to compute $f_{\text{LR}}(\boldsymbol{w} + \boldsymbol{d})$ (in line 5 of Algorithm 1). However, large-memory transfers between the host and device are expensive, especially when either the host or device are waiting idle for the transaction to complete (steps to conceal transfer latency are discussed in [23]).

Furthermore, all other major operations in Algorithm 1 are locked in the same manner as the previous bottleneck example: the trust region update (lines 6-10) can not proceed without the value of $f_{\text{LR}}(\boldsymbol{w} + \boldsymbol{d})$, and, without either the updated $\boldsymbol{w}$ or trust region, operations for the next iteration's truncated Newton step (line 4) are unable to run concurrently in an attempt to conceal transfer latency. Clearly, this pattern of variable access is suboptimal for GPU use (best evidenced by *TRON*-LR-GPU$^0$'s performance in Section 6).

## 4.2 Decoupling Dependencies to Maximize Host and Device Parallelism

To optimally use the GPU, we must first decouple the sequential dependency of variables discussed in Section 4.1. Recall that, for $\nabla f_{\text{LR}}(\boldsymbol{w})$, the vector $\hat{\boldsymbol{z}}$ is such that $\hat{\boldsymbol{z}}_i = (h(y_i \boldsymbol{z}_i) - 1)y_i$. To decrease sequential dependencies on the computational bottleneck $z = X^T(\boldsymbol{w} + \boldsymbol{d})$, we first note that calculation of $f_{\text{LR}}(\boldsymbol{w} + \boldsymbol{d})$ always precedes $\nabla f_{\text{LR}}(\boldsymbol{w} + \boldsymbol{d})$. Thus, to decouple gradient variables, once $z$ is calculated on the device, we prepare all device-side variables needed to compute $X^T\hat{\boldsymbol{z}}$ in the event that $\sigma > \sigma_0$. Specifically, after computing $z = X^T(\boldsymbol{w} + \boldsymbol{d})$ on the device, we use a custom CUDA kernel to compute $\hat{\boldsymbol{z}}$ followed by a Thrust reduction to compute $f_{\text{LR}}(\boldsymbol{w} + \boldsymbol{d}) = \frac{1}{2}\|\boldsymbol{w} + \boldsymbol{d}\|_2^2 + C\sum_{i=1}^l \log(1 + e^{-y_i \boldsymbol{z}_i})$ (note that the scalar output of the reduction, i.e. $f_{\text{LR}}(\boldsymbol{w} + \boldsymbol{d})$, is immediately available to the host). The computation of $\hat{\boldsymbol{z}}$ is massively parallelizable, so the grid-stride loop in the custom kernel is extremely efficient. Thus, if $\sigma > \sigma_0$, the variable $\hat{\boldsymbol{z}}$ is already in device memory and the gradient is quickly calculated using cuSPARSE on the device as $\nabla f_{\text{LR}}(\boldsymbol{w} + \boldsymbol{d}) = \boldsymbol{w} + \boldsymbol{d} + X^T\hat{\boldsymbol{z}}$. Finally, $\nabla f_{\text{LR}}(\boldsymbol{w} + \boldsymbol{d})$ is transferred from device to host, which is notably efficient when $l \gg n$ (i.e., the optimal setting for primal learning).

This set of operations accomplishes several optimizations simultaneously:

- **Decoupling dependencies, avoiding large transfers**: $\boldsymbol{z}$ and $\hat{\boldsymbol{z}}$ are completely decoupled of any dependency for host-side computation, thanks to the custom reduction and kernel. This saves several large transfers of $\boldsymbol{z}, \hat{\boldsymbol{z}}$ from (and to) the device, and avoids the need to conceal transfer latency.

- **Coalesced memory**: the device performs optimally as all operations allow memory coalescing.

- **Device saturation**: an uninterrupted series of intensive computation is performed on the device (i.e., no device-side stalls due to host dependencies).

- **Host and device parallelism**: the complete decoupling of $\boldsymbol{z}, \hat{\boldsymbol{z}}$ allows more independent operations to be run on the host while the device runs concurrently.

We complete the total GPU optimization of *TRON*-LR by speeding up the remaining bottleneck, the Hessian-vector product $\nabla^2 f_{\text{LR}}(\boldsymbol{w})\boldsymbol{v}$. As with the previous optimizations, device variables are maximally decoupled from host-side dependencies, while using device-side functions which allow

| *TRON*-SVM-GPU | *TRON*-SVM-MIX |
|---|---|
| $\boldsymbol{z} = X\boldsymbol{w}$ is calculated and stored on the device. | $\boldsymbol{z} = X\boldsymbol{w}$ is calculated and stored on the device, then transferred to the host. |
| $I = \{i : y_i \boldsymbol{z}_i < 1\}$ is calculated on the device, then $f_{\text{L2}}(\boldsymbol{w}) = \frac{1}{2}\|\boldsymbol{w}\|_2^2 + C\sum_{i=1}^{l}(1 - y_i\boldsymbol{z}_i > 0)^2$ is computed on the device while the host runs independent, sequential operations. | $I$ is calculated on the device, then transferred to the host. The device-side computation of $f_{\text{L2}}(\boldsymbol{w})$ is run concurrently with this transfer. |
| On the device, $\hat{\boldsymbol{z}} = (\boldsymbol{z}_I - \boldsymbol{y}_I)$ and $\hat{X} = X_{I,:}$ are computed. The gradient $\nabla f_{\text{L2}}(\boldsymbol{w}) = \boldsymbol{w} + 2C\hat{X}^T\hat{\boldsymbol{z}}$ is then computed and transferred to the host. | With $I$ and $z$ on the host, $\nabla f_{\text{L2}}(\boldsymbol{w})$ is computed using multithreading. |
| The Hessian-product is computed on the device as $\nabla^2 f_{\text{L2}}(\boldsymbol{w})\boldsymbol{v} = \boldsymbol{v} + 2C\hat{X}^T(\hat{X}\boldsymbol{v})$ and transferred to the host. | Using multithreading, the Hessian-product is calculated on the host as $\nabla^2 f_{\text{L2}}(\boldsymbol{w})\boldsymbol{v} = \boldsymbol{v} + 2CX_{I,:}^T(X_{I,:}\boldsymbol{v}) = \boldsymbol{v} + 2C\sum_{i\in I}(\boldsymbol{x}_i^T\boldsymbol{v})\boldsymbol{x}_i$. |

Table 2: Major operations of the *TRON*-SVM solvers designed for GPU compute.

peak performance. Further details are in [23], including a comprehensive summary of the previously described *TRON*-LR GPU-optimizations.

**Decreasing runtimes via mixed-architecture speedups.** While computations remain which may be accelerated using the same GPU-optimization principles, allocating additional device vectors becomes problematic for large-scale datasets and current GPU memory ranges. Thus, in addition to the previously described GPU optimizations, we accelerate remaining bottleneck areas using multithreading. In particular, multithreading using OpenMP is used to accelerate vector-matrix-vector multiplications in the conjugate gradient procedure (previously optimized using loop unrolling) and application of the preconditioner matrix [25] (which is jointly sped up using existing device-side computation during the Hessian-vector product optimizations).

## 5  Accelerating *TRON*-SVM training using GPUs

Focusing on speeding up SVM learning in the state-of-the-art software Percolator [33]–which uses dense feature vectors to analyze large-scale proteomics datasets–the GPU-optimization principles from Section 4.2 are applied to *TRON*-SVM: device-side variables are decoupled from dependent host-side computations, necessary transfers are run asynchronously in parallel with the maximum number of host/device operations, and linear algebra operations are extensively and efficiently carried out using cuBLAS. However, speed ups in *TRON*-SVM possess a key difficulty for GPU computation; for $\boldsymbol{z} = X^T w$, the active set $I \equiv \{i | 1 - y_i\boldsymbol{z}_i > 0\}$ is recomputed every iteration. Thus, computation of both $\nabla f_{\text{L2}}(\boldsymbol{w})$ and $\nabla^2 f_{\text{L2}}(\boldsymbol{w})$ require the submatrix $X_{I,:}$ (as seen in Table 1).

While accessing $X_{I,:}$ is naturally supported through shared-memory random access for multithreaded speedups (e.g., in [22], which used OpenMP to speed up the *TRON*-SVM quantities in Table 1 within Percolator), the non-contiguous nature of this operation leads to misaligned (i.e., not coalesced) device memory which prevents optimal GPU use. Furthermore, as noted in Section 4.2, large memory transfers between host and device are expensive, hindering approaches where $I$ is first computed then a randomly accessed submatrix is created on the host and transferred to the device.

To overcome this challenge, we first make use of the insight that, prior to computing $f_{\text{L2}}(\boldsymbol{w})$, the active set $I$ may be computed and stored entirely on the device. With $I$ on the device, the submatrix $X_{I,:}$ may be efficiently computed *within device memory*. Computing $I$ and $X_{I,:}$ on the device entirely decouples these variables from host-side compute and accomplishes all simultaneous optimizations listed in Section 4.2. The major operations of the resulting GPU-optimized solver, called *TRON*-SVM-GPU, are listed in Table 2.

**Decreasing GPU-memory utilization via mixed-architecture speedups.** In *TRON*-SVM-GPU, the device memory required to decouple $X_{I,:}$ from host-side compute proves prohibitive for extremely large-scale proteomics datasets. To remedy this, the mixed-architecture solver, *TRON*-SVM-MIX, utilizes the GPU for heavy lifting before using multithreading for efficient random access to $X_{I,:}$ (after $I$ is computed) during *TRON*'s conjugate gradient procedure. Thus, *TRON*-SVM-MIX utilizes much less GPU memory than *TRON*-SVM-GPU, at the expense of some speed due to fewer operations being run on the device. The major operations of *TRON*-SVM-MIX are listed in Table 2.

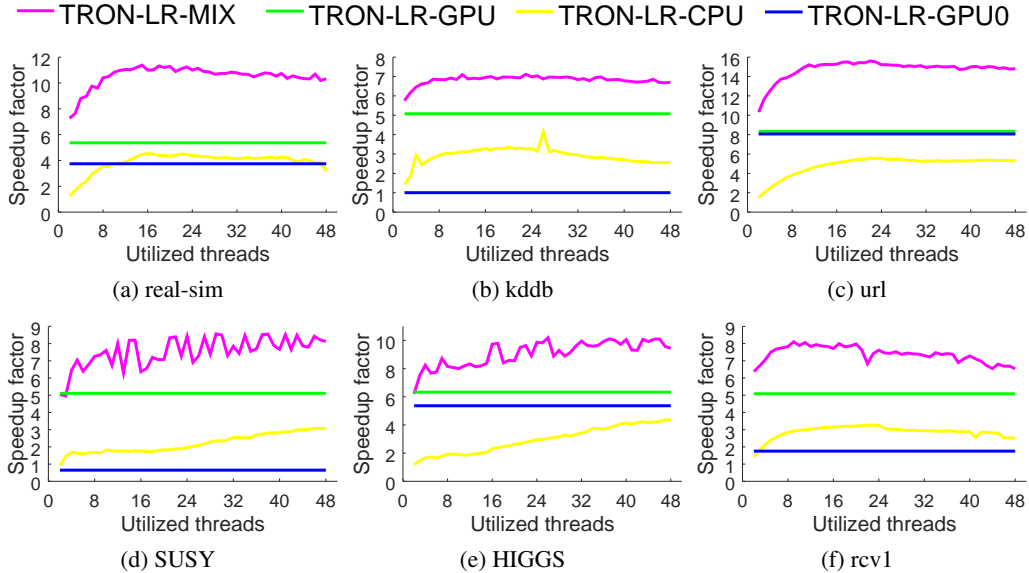

Figure 1: Factor of speedup for several optimized versions of *TRON*-LR in LIBLINEAR. The $x$-axis displays the number of threads used for multithreaded methods. The $y$-axis denotes the multiplicative factor of training speedup for each method relative to the single-threaded version of *TRON*-LR in the standard LIBLINEAR. Training times were measured within LIBLINEAR as the time elapsed calling `tron_obj.tron()`. As is standard practice, wallclock times were measured as the minimum reported times over ten runs.

## 6 Results and Discussion

All experiments were run on a dual Intel Xeon Gold 5118 compute node with 48 computational threads, an NVIDIA Tesla V100 GPU, and 768 GB of memory.

**Speedups for sparse features**. The *TRON*-LR GPU-optimized and mixed-architecture solvers (described in Section 4.2) are referred to as *TRON*-LR-GPU and *TRON*-LR-MIX, respectively. *TRON*-LR-GPU, *TRON*-LR-MIX, and *TRON*-LR-GPU$^0$ were all developed based on LIBLINEAR v2.30. Single-threaded LIBLINEAR tests were run using v2.30. The multithread-optimized version of *TRON*-LR described in [36], referred to herein as *TRON*-LR-CPU, was tested using multi-core LIBLINEAR v2.30. All single-threaded *TRON*-LR implementations (i.e., *TRON*-LR-GPU$^0$, *TRON*-LR-GPU, and the single-threaded optimized version of *TRON*-LR in standard LIBLINEAR) were run with the same command line parameters: `-c 4 -e 0.1 -s 0`. Multithreaded implementations were run with the additional flag `-nr i`, specifying the use of $i$ compute threads. As is standard practice, wallclock times were measured as the minimum reported times over ten runs. Training times were measured within LIBLINEAR as the time elapsed calling `tron_obj.tron()`. Six datasets of varying statistics (i.e., number of features, instances, and nonzero elements) were downloaded from https://www.csie.ntu.edu.tw/~cjlin/libsvmtools/datasets/ and used to benchmark the *TRON*-LR solvers (statistics for each dataset are listed in [23]).

The speedups for all methods are displayed in Figure 1. *TRON*-LR-GPU significantly outperforms the multithread-optimized *TRON*-LR-CPU and the direct GPU implementation, *TRON*-LR-GPU$^0$, across all datasets and threads. The mixed-architecture *TRON*-LR-MIX further improves upon *TRON*-LR-GPU performance in each dataset for all threads used, leading to over tenfold speedups in training time on half of the presented datasets. We note that, due to thread scheduling overhead, multithreaded methods experience diminished performance for large numbers of threads in Figures 1a,1b,1f. However, the presented GPU optimizations consistently provide the best speedups when both multithreading is not used and when multithreading is overutilized.

**Speedups for dense features.** The *TRON* GPU solvers described in Section 5–the GPU-optimized *TRON*-SVM-GPU and the mixed-architecture *TRON*-SVM-MIX–are compared against the multithread-optimized versions of *TRON* (referred to as *TRON*-SVM-CPU) and *L2-SVM-MFN* from [22]. The methods are tested using two extremely large datasets. The first dataset, referred to as the Kim dataset, is a larger version of the benchmark dataset used in [22], consisting of 23,330,311 PSMs (i.e., proteomics data instances, described in 2.1). The second dataset, referred to as the

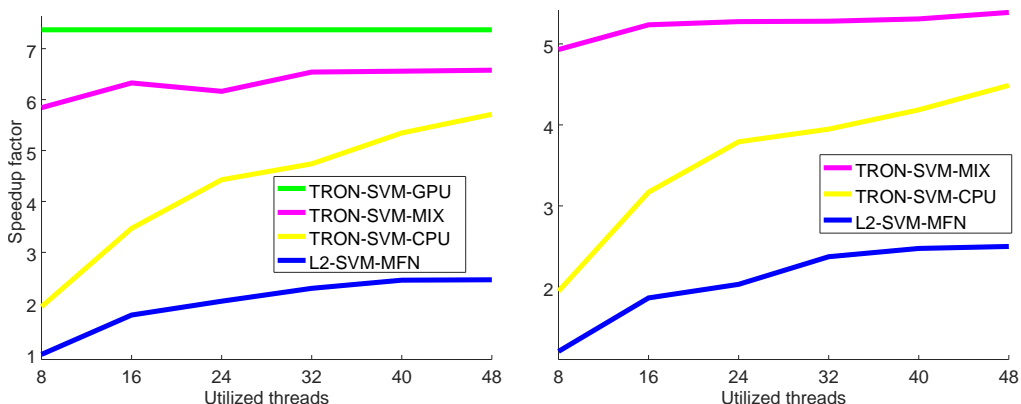

(a) SVM speedups for a large, dense proteomics dataset containing 23,330,311 PSMs.

(b) SVM speedups for a massive dense dataset containing 215,282,771 PSMs, too large to be analyzed by "TRON-SVM-GPU."

Figure 2: Factor of speedup for SVM learning in Percolator for dense large- and massive-scale datasets. Speedup factor is calculated as the original Percolator SVM learning time divided by the sped up learning time. The x-axis displays the number of threads utilized by multithreaded methods "L2-SVM-MFN," "TRON-SVM-CPU," and "TRON-SVM-MIX."

Wilhelm dataset, was collected from a map of the human proteome [51] and contains 215,282,771 PSMs. All multithreaded solvers were tested using 8, 16, 24, 32, 40, and 48 threads. As in [22], to effectively measure the runtime of multithreaded methods without excess thread-scheduling overhead, parallelization of Percolator's outermost cross-validation was disabled.

Reported runtimes are the minimum wall-clock times measured over five runs for the Kim dataset and three runs for the Wilhelm dataset. The original Percolator SVM learning runtimes (collected using Percolator v3.04.0) were 14.4 hours and 4.4. days for the Kim and Wilhelm datasets, respectively. Speedups for both datasets are illustrated in Figure 2. For the Kim dataset, speedup results for all discussed methods are illustrated in Figure 2a. For the Wilhelm dataset, total Tesla V100 memory (16 GB) is exceeded for *TRON*-SVM-GPU. However, the reduced memory requirements of *TRON*-SVM-MIX allow GPU speedups for this massive dataset (illustrated in Figure 2b).

Both GPU solvers greatly accelerate Percolator SVM learning while dominating previously proposed multithreaded speedups. For the Kim dataset, *TRON*-SVM-MIX and *TRON*-SVM-GPU achieve 6.6 and 7.4 fold speedups, respectively, over Percolator's current SVM learning engine. For the Wilhelm dataset, *TRON*-SVM-MIX achieves a 5.4 fold speedup while being notably efficient using few system threads–with at most 16 threads, *TRON*-SVM-MIX improves the average training time of *TRON*-SVM-CPU and *L2-SVM-MFN* by 50% and 70%, respectively. Together, these two solvers present versatile trade-offs for different compute environments; when the dataset does not exceed the GPU memory, *TRON*-SVM-GPU offers superior performance. However, when onboard GPU memory is limited, a small portion of speed may be traded for much less memory consumption by using *TRON*-SVM-MIX. Furthermore, when the number of computational threads is also limited, *TRON*-SVM-MIX offers significantly better (and more stable) performance at low numbers of utilized threads compared to the purely multithreaded solvers *TRON*-SVM-CPU and *L2-SVM-MFN*.

# 7    Conclusions and Future Work

In this work, we've shown that by using general GPU-optimization principles, excellent speedups may be enjoyed by algorithms which natively resist GPU optimization. For the widely used *TRON* algorithm, we've presented several GPU-optimized solvers for both sparse and dense feature sets of $L_2$-regularized primal problems. Using a single GPU, these solvers were shown to dominate recently proposed speedups for logistic regression (within LIBLINEAR) and SVM classification for state-of-the-art proteomics analysis (within Percolator). Furthermore, for sparse features, we've shown how multithreading may compliment GPU optimizations and, for memory-restrictive dense features, how multithreading may relieve device-memory requirements while allowing substantial GPU speedups. The former optimizations achieve over an order-of-magnitude speedup on half of the

presented datasets (and an average 9.3 fold speedup on all datasets), while the latter optimizations decrease massive-scale biological analysis time from 4.4 days down to just 19.7 hours.

There are significant avenues for future work. We plan to extend GPU-optimized *TRON* implementations to use general gradient and Hessian-vector product information computed in auotomatic differentiation [5] packages such as PyTorch [44] and TensorFlow [1], which utilize second-order primal solvers (such as *L-BFGS*) to optimize losses while relying on GPU compute for optimal performance. Furthermore, we plan to apply the presented GPU-optimization principles to speed up other fast machine learning solvers [27, 2, 54, 35, 31, 32] which, like *TRON*, are natively designed to rely on sequential dependencies of variables.

## Broader Impact

This paper solely focuses on speeding up machine learning software, and thus impacts machine learning packages or applications which use either the included software or the paper's GPU optimization principles (to speed up an algorithm not discussed). Benefits include faster software, with specific applications including real-time classification for self-driving cars [53], flagging credit card fraud [12], water monitoring to preserve ecosystems in maritime and archipelagic countries [3]), etc. Machine learning companies/researchers/practitioners who do not use GPU resources may be put at a disadvantage from this research, but any advantage/disadvantage is defined solely in terms of training time speed.

**Acknowledgments**: This work was supported by the National Center for Advancing Translational Sciences (NCATS), National Institutes of Health, through grant UL1 TR001860 and a GPU donation from the NVIDIA Corporation.

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
