[Supplementary Material]

# GPU-Accelerated Primal Learning for Extremely Fast Large-Scale Classification: Supplementary Material

**John T. Halloran**
Department of Public Health Sciences
University of California, Davis
jthalloran@ucdavis.edu

**David M. Rocke**
Department of Public Health Sciences
University of California, Davis
dmrocke@ucdavis.edu

## 1 GPU speedups training a logistic regression classifier in PyTorch

A binary logistic regression classifier was implemented in PyTorch (v1.4.0 ) and trained over the rcv1 dataset to illustrate the speed ups possible using a GPU (Nvidia Tesla V100) versus only multithreading (24 CPU threads using an Intel Xeon Gold 5118). Speedups were tested for both batch gradient descent (with a 0.001 learning rate) and *L-BFGS*. The rcv1 dataset was downloaded from https://www.csie.ntu.edu.tw/~cjlin/libsvmtools/datasets/binary/rcv1_train.binary.bz2. Gradient descent converged after 3,000 iterations and *L-BFGS* converged after 100 iterations. For reference, a logistic regression classifier was trained using single-threaded *TRON* (as implemented in scikit-learn v0.20.4). All code is available in pyTorchLogisticRegression_rcv1.py.

| Solver | CPU training time (s) | GPU training time (s) | GPU Speedup |
|---|---|---|---|
| Gradient descent | 395.58 | 27.05 | 14.63 |
| *L-BFGS* | 40.56 | 3.1 | 13.08 |
| *TRON* (scikit-learn) | 0.29 | – | – |

Table 1: Logistic regression training times, measured in seconds, for the rcv1 dataset. Gradient descent and *L-BFGS* solvers are implemented in PyTorch, and single-threaded *TRON* is implemented in scikit-learn.

## 2 Derivation of *TRON* Hessian-vector products

Consider feature vectors $\boldsymbol{x}_i \in \mathbb{R}^n, i = 1,\ldots,l$ and label vector $\boldsymbol{y} \in \{-1,1\}^l$, and let $X = [\boldsymbol{x}_1 \ldots \boldsymbol{x}_l]^T$ be the feature matrix. For vectors, index-set subscripts denote subvectors and for matrices, pairs of index-set subscripts denote submatrices. Let $\boldsymbol{1}$ denote the indicator function.

The general $L_2$-regularized objective, which we wish to minimize w.r.t. $\boldsymbol{w}$, is

$$f(\boldsymbol{w}) = \frac{1}{2}\boldsymbol{w}^T\boldsymbol{w} + C\sum_{i=1}^{l} \ell(\boldsymbol{w};\boldsymbol{x}_i,y_i), \tag{1}$$

where $\frac{1}{2}\boldsymbol{w}^T\boldsymbol{w}$ is the regularization term, $C > 0$ is a regularization hyperparameter, and $\ell(\boldsymbol{w};\boldsymbol{x}_i,y_i)$ is a loss function. When $\ell(\boldsymbol{w};\boldsymbol{x}_i,y_i) = \log(1 + \exp{(-y_i\boldsymbol{w}^T\boldsymbol{x}_i)})$, commonly referred to as the logistic loss, minimizing Equation 1 corresponds to learning a classifier using logistic regression. Similarly, minimizing Equation 1 when $\ell(\boldsymbol{w};\boldsymbol{x}_i,y_i) = (\max(0, 1 - y_i\boldsymbol{w}^T\boldsymbol{x}_i))^2$, commonly referred to as the L2-SVM or quadratic SVM loss, corresponds to learning a linear SVM classifier. The logistic loss results in an objective function that is twice differentiable and the L2-SVM loss yields a differentiable

objective (unlike the hinge loss) with a generalized Hessian [2]). We denote Equation 1 under the logistic loss as $f_{\text{LR}}$ and, under the $L_2$-SVM loss, as $f_{\text{L2}}$.

*TRON* is detailed in Algorithm 1. At each iteration, given the current parameters $w$ and trust region interval $\Delta$, TRON considers the following quadratic approximation to $f(w + d) - f(w)$,

$$q(d) = \nabla f(w)^T d + \frac{1}{2} d^T \nabla^2 f(w) d. \tag{2}$$

A truncated Newton step, confined in the trust region, is then found by solving

$$\min_{d} q(d) \quad \text{s.t.} \ \|d\|_2 \leq \Delta. \tag{3}$$

If $q(d)$ is close to $f(w + d) - f(w)$, $w$ is updated to $w + d$ and the trust region interval is increased for the subsequent iteration. Otherwise, $w$ remains unchanged and the trust region interval is shrunk.

---

**Algorithm 1** The *TRON* algorithm

---

1: Given $w$, $\Delta$, and $\sigma_0$
2: Calculate $f(w)$                                    // *Critically depends on $z = X^T w$*
3: **while** Not converged **do**
4:     Find $d = \operatorname{argmin}_v q(v)$ s.t. $\|v\|_2 \leq \Delta$.     // *Critically depends on $\nabla f(w)$, $\nabla^2 f(w) v$*
5:     Calculate $f(w + d)$, $\sigma = \frac{f(w+d) - f(w)}{q(d)}$       // *Critically depends on $z = X^T(w + d)$*
6:     **if** $\sigma > \sigma_0$ **then**
7:         $w \leftarrow w + d$, increase trust region $\Delta$.
8:     **else**
9:         Shrink $\Delta$.
10:     **end if**
11: **end while**

---

Note that the function evaluation $f(w)$ must be computed for each new iteration, as well as the gradient and the Hessian for Equation 2. However, Equation 2 involves only a Hessian-vector product, computation of which circumvents loading the entire Hessian into memory. For the logistic loss, we have

$$\nabla f_{\text{LR}}(w) = w + C \sum_{i=1}^{l} (h(y_i w^T x_i) - 1) y_i x_i, \tag{4}$$

where $h(y_i w^T x_i) = (1 + e^{-y_i w^T x_i})^{-1}$. For the L2-SVM loss, we have

$$\nabla f_{\text{L2}}(w) = w + 2C \hat{X}^T \hat{z} = w + 2C X_{I,:}^T (X_{I,:} w - y_I), \tag{5}$$

where $I \equiv \{i | 1 - y_i w^T x_i > 0\}$ is an index set and and the operator : denotes all elements along the corresponding dimension (i.e., all columns in this case). Thus, $X_{I,:}$ is the submatrix of all $X$ rows the indices of which are in $I$.

Equation 3 involves only a single Hessian-vector product, the structure of which is exploited to avoid loading the entire Hessian into memory. For the logistic loss, we have

$$\nabla^2 f_{\text{LR}}(w) = \mathcal{I} + C X^T D X, \tag{6}$$

where $D$ is a diagonal matrix with elements $D_{i,i} = h(y_i w^T x_i)(1 - h(y_i w^T x_i))$. Thus, for a vector $v$, the Hessian-vector product is efficiently computed as $\nabla^2 f_{\text{LR}}(w) v = v + C X^T (D(X v))$. For the L2-SVM loss, we have

$$\nabla^2 f_{\text{L2}}(w) = \mathcal{I} + 2C X^T D X = \mathcal{I} + 2C X_{I,:}^T X_{I,:}, \tag{7}$$

where $D$ is a diagonal matrix with elements $D_{i,i} = \mathbf{1}_{i \in I}$. The Hessian-vector product is thus efficiently computed as $\nabla^2 f_{\text{L2}}(w) v = v + 2C X_{I,:}^T (X_{I,:} v)$.

## 3 Concealing large-memory transfer latency between the host and device

To optimally conceal device-to-host transfer latency while maximizing host and device parallelism, it is necessary to:

(a) Add all dependent device-functions involving the data to be sent to an asynchronous device stream, $s$,

(b) add the transfer of the data from device-to-host to $s$,

(c) run independent host and/or device operations,

(d) synchronize $s$ just prior to running a dependent operation on the host.

(e) Note that if the dependent data needed from the device on the host is a scalar, it may be returned without latency.

The other direction is slightly different. To optimally conceal host-to-device transfer latency while maximizing host and device parallelism, it is necessary to:

(a) launch the transfer on a device stream as soon as the data is available,

(b) add all dependent device-functions involving the data being sent to the device stream.

It is easy to see that algorithms with many sequential dependencies are at odds with these principles (they reveal transfer latency while minimizing host/device parallelism).

## 4  Optimization of *TRON*-LR Hessian-vector products for GPUs

We complete the total GPU-optimization of *TRON*-LR by considering the remaining bottleneck, the Hessian-vector product $\nabla^2 f_{\text{LR}}(\boldsymbol{w})\boldsymbol{v} = \mathcal{I} + CX^T(D(X\boldsymbol{v}))$. As with the previous optimizations, device variables are maximally decoupled from host-side dependencies, while using device-side functions which allow peak performance. In particular, we compute the diagonal matrix $D$ in the same custom CUDA kernel used to compute $\hat{\boldsymbol{z}}$ (where $D_{i,i} = h(y_i\boldsymbol{z}_i)(1 - h(y_i\boldsymbol{z}_i))$. $D$ is also used in later host computations (for preconditioning [1]), so $D$ is immediately transferred from device to host on an asynchronous device stream (the stream is synchronized just prior to host-variable use).

The candidate Newton step $\boldsymbol{v}$ (which is only of dimension $n$) is transferred from device to host on an asynchronous stream, and the following decompositions of $\nabla^2 f_{\text{LR}}(\boldsymbol{w})\boldsymbol{v}$ are added to this same stream: $\boldsymbol{a}_0 = X\boldsymbol{v}, \boldsymbol{a}_1 = D\boldsymbol{a}_0, \boldsymbol{a}_2 = CX^T\boldsymbol{a}_1$. $\boldsymbol{a}_0$ and $\boldsymbol{a}_2$ are computed using cuSPARSE, while $\boldsymbol{a}_1$ is computed using a custom kernel for element-wise multiplication along $D$'s diagonal. $\nabla^2 f_{\text{LR}}(\boldsymbol{w})\boldsymbol{v}$ is then transferred from host to device. However, the rest of the conjugate procedure is sequentially dependent on the dot-product $\boldsymbol{v}^T\nabla^2 f_{\text{LR}}(\boldsymbol{w})\boldsymbol{v}$. In order to relieve this dependence while the $\nabla^2 f_{\text{LR}}(\boldsymbol{w})\boldsymbol{v}$ transfers from device to host, $\boldsymbol{v}^T\nabla^2 f_{\text{LR}}(\boldsymbol{w})\boldsymbol{v}$ is computed on the device and the resulting scalar is available immediately to the host.

## 5  Summary of major *TRON*-LR-GPU operations

The following summarizes the major operations of the GPU-optimized *TRON* logistic regression solver, *TRON*-LR-GPU, as described in the main paper and herein. For each set of operations, the original lines from Algorithm 1 being optimized are listed in red.

- $\boldsymbol{z} = X\boldsymbol{w}$ is calculated and stored on the device (lines 2 and 5).

- The vectors $\boldsymbol{\alpha}$, $\hat{\boldsymbol{z}}$ and diagonal matrix $D$ are calculated on the device, such that $\boldsymbol{\alpha} = \log(1/h(y_i\boldsymbol{z}_i))$, $\hat{\boldsymbol{z}}_i = (h(y_i\boldsymbol{z}_i) - 1)y_i$, and $D_{i,i} = h(y_i\boldsymbol{z}_i)(1 - h(y_i\boldsymbol{z}_i))$, where $h(y_i\boldsymbol{z}_i) = (1 + e^{-y_i\boldsymbol{z}_i})^{-1}$ (lines 2 and 5). $D$ is asynchronously transferred back to the host for future preconditioning computations.

- With $\boldsymbol{\alpha}$ in device memory, the objective $f_{\text{LR}}(\boldsymbol{w}) = \frac{1}{2}\boldsymbol{w}^T\boldsymbol{w} + C\sum_{i=1}^{l}\log(1 + \exp(-y_i\boldsymbol{z}_i)) = \frac{1}{2}\boldsymbol{w}^T\boldsymbol{w} + C\sum_{i=1}^{l}\boldsymbol{\alpha}$ is computed (lines 2 and 5).

- With $\hat{\boldsymbol{z}}$ in device memory, the gradient $\nabla f_{\text{LR}}(\boldsymbol{w}) = \boldsymbol{w} + X^T\hat{\boldsymbol{z}}$ is computed and transferred asynchronously back to the host (line 7).

- While all the above device-side quantities are being computed, the host runs independent, sequential operations concurrently, synchronizing the transfer streams for $D$ and $\nabla f_{\text{LR}}(\boldsymbol{w})$ just prior to host-side use (lines 7 and 4, respectively).

- The Hessian-product is computed on the device as $\nabla^2 f_{\text{LR}}(\boldsymbol{w})\boldsymbol{v} = \boldsymbol{v} + CX^T(D(X\boldsymbol{v}))$. Subsequently, the vector-Hessian-vector product $\boldsymbol{v}^T\nabla^2 f_{\text{LR}}(\boldsymbol{w})\boldsymbol{v}$ is computed on the device and the resulting scalar is immediately available to the host (line 4).

# 6 Benchmark Dataset Statistics

| Dataset | #instances | #features | #nonzeros |
|---|---|---|---|
| rcv1 | 20,242 | 47,236 | 1,498,952 |
| SUSY | 5,000,000 | 18 | 88,938,127 |
| HIGGS | 11,000,000 | 28 | 283,685,620 |
| KDD2010-b | 19,264,097 | 29,890,095 | 566,345,888 |
| url | 2,396,130 | 3,231,961 | 277,058,644 |
| real-sim | 72,309 | 20,958 | 3,709,083 |
| Kim | 23,330,311 | 18 | 419,945,598 |
| Wilhelm | 215,282,771 | 18 | 3,875,089,878 |

Table 2: Sparse and dense benchmark dataset statistics for *TRON*-LR and *TRON*-SVM, respectively.