[Reviews · NeurIPS 2020]

Review 1

Summary and Contributions: This paper proposes optimizations to introduce GPUs to further improve the speedup achieved by the TRON algorithm. Specifically, the authors propose to increase parallel processing on CPUs and GPUs, by decoupling the sequential dependencies of variables, minimizing the frequency of large memory transfers between CPU and GPU. The paper also proposes to use multi-threading for cases with large-scale datasets that pose challenges to the limited memory of GPUs. ===== After the author response: I read the author response, and I am happy with their answers to my questions in their response.

Strengths: This paper tackles an important problem of offloading computation to GPUs and utilizing their computation capabilities, proposes optimizations that are intuitive, explains them well, and evaluates their effectiveness.

Weaknesses: Two key missing point I felt were the lack of GPU-offloading related work, and a discussion about more general applicability of the proposed optimizations.

Correctness: I believe the described optimizations and evaluation methodology is correct; it would be great to understand how generalizable the proposed optimizations for GPU offloading are (for other algorithms apart from LR and SVM).

Clarity: The paper reads well, motivates the problem well, and describes the optimizations well too. A more detailed description of the custom kernel and the reduction used for the optimization in section 4.2 would help. Nit: Please consider using the colors consistently for the legends across graphs, for improved readability. For instance, fig 1 uses pink for MIX, while currently fig 2 uses pink for the GPU model.

Relation to Prior Work: It will be great if the authors could add details about existing work related to offloading computation to GPU in this and related context. If there is no such work that is directly relevant, it will also help if you could say that explicitly.

Reproducibility: Yes

Additional Feedback: Section 4.2 mentions the use of a custom cuda kernel and a reduction. This seems to be the key to achieving this optimization of decoupling the dependencies between host and device. Please consider elaborating a bit more on this, explaining how the use of custom kernel and the reduction presented here leads to these optimizations. It wasn’t entirely clear to me what the difference is between the proposed mixed architecture using GPU/device and multi-threading from the state-of-the-art, which from I understand in section 2 also used GPU for limited computation, and uses CPUs for the rest. I might be missing something, but it would be great if this difference is explicitly mentioned. ----- After rebuttal: Thank you for submitting a detailed response, it clarified my concerns well.


Review 2

Summary and Contributions: Trust Region Newton Algorithm (TRON) is the most efficient solver for L2 regularized primal problems e.g. Logistic regression (LR) and Linear Support Vector Machines (SVMs). Due to the complex and sequential nature of this algo., its past performance boosts have largely been driven by shared memory multi-core systems. This paper demonstrates significant speedups in the training time of TRON solver compared to multithreaded implementations by using GPU specific optimization principles. The authors apply specific optimizations on sparse representation (LR training) and dense representation problems (SVM training) to generate significant speedups in their training time using GPUs. Specifically, for sparse feature representation datasets and LR loss function, the authors prescribe optimizations that minimize sequential dependence of CPU/GPU execution on each other by assuming all conditional branches evaluate in favor of the high-compute operations that can be run pre-emptively on the GPU. Additionally, the data flow is customized to select smallest possible transfers between the CPU and GPU. These optimizations are extended to dense feature representation datasets and SVM loss function accounting for the unique dataflow patterns of this loss function. More importantly for the dense feature representation datasets, the authors prescribe a different scheduling strategy and dataflow that distributes the computation between GPU (evaluating loss function for given weights) and multi-threaded cores (Hessian-product computation) which plays to the strengths of these resources. This set of optimizations allows the incorporation of GPU along with multi-threaded cores to speedup massive dataset (~100million data instances) training using TRON. Such training is infeasible on a GPU only optimized scheduling scheme due to limited GPU memory. The paper claims up to 10x improvement in training time for GPU optimized TRON solver over multi-core optimized TRON solver on sparse representation datasets (using LR loss function). It also claims a ~5x speedup for dense representation proteomics dataset (215 million data inst.) using TRON-SVM solver.

Strengths: The authors present a first attempt at optimizing the TRON solver for utilizing GPU and demonstrate significant speedups in the training times for sparse and dense feature representation datasets using LR and SVM loss functions respectively. Additionally, the mixed (GPU + multi-threaded CPU) hardware resource optimizations prescribed by the paper show strong speedups in the training time for massive (100s of million data instances) datasets with the example of a realistic problem of SVM classification in the proteomics analysis pipeline. These optimization principles will allow the use of GPU (when available) instead of highly multi-threaded multi-core systems (might not have enough threads) in speeding primal learning problems. The theoretical formulation and breakdown of the key steps involved in solving LR/SVM loss function training problems is clean and commendable. The paper provides ample details related to the experimental conditions (CPU/GPU specs) and uses a standard library (LIBLINEAR) to test their optimizations. These details will be helpful in reproducing the results presented in the paper.

Weaknesses: Based on the data presented in Figure 1, the authors claim a speedup of ~10x in half of the six (sparse representation) datasets analyzed. This claim does not seem to be warranted by the presented data. Considering the TRON-LR-CPU (baseline multi-threaded CPU optimized TRON) as reference, TRON-LR-MIX comes close to the 10x speedup mentioned only for very few (1-2) threads. Whereas, for reasonably higher number of threads (~16-24) most of these analyzed datasets show only ~3x speedup of TRON-LR-MIX over TRON-LR-CPU. I think that while the optimizations proposed in this paper enable first ever GPU based speedups for TRON with the use of few (1-2) threads, it is unfair to quantify that speedup as a benefit over state-of-the-art multi-threaded TRON solver. Since the proposed TRON-LR-MIX solution is run on a powerful NVIDIA Tesla V100 GPU (costs ~$10,000), it’s only fair to assume a significantly higher number of threads on a TRON-LR-CPU baseline. Keeping this in mind, the speedup claim by the paper for sparse datasets should be updated. Author Feedback Response: Thanks to the authors for clarifying my concern, the authors do clearly specify on line 60-61 that the speedup measured for the proposed GPU and MIX system optimizations compared to multi-threaded optimized TRON solvers in LIBLINEAR is ~65.2% (~3x) which aligns with my expectation. Further, considering the TRON-LR-GPU0 (default LIBLINEAR GPU execution with no optimization) as reference. Few datasets show small to negligible improvement by TRON-LR-GPU over its unoptimized counterpart (e.g. url). These datasets are interesting in the sense that only the combination of GPU and multi-threaded resources (~16-24 threads) can help speedup these datasets significantly. It would be beneficial if the authors could identify the unique characteristics of these datasets that render their training times unaffected by only GPU based optimizations Figure 2 describes the speedups achieved through GPU and MIX optimization strategies over multi-threaded CPU baseline. The discussion on this figure claims similar speedup benefits as Figure 1 using few (1-2, Kim dataset and 8, Wilhelm dataset) threads as baseline which isn’t fair. With reasonably larger number of threads, the speedup claims should be more modest. Although the authors utilize the maximum available threads on their selected CPU platform there exist other platforms with even higher number of threads (AMD Ryzen Threadripper can use up to 128 threads). It would be interesting to see at what thread count the benefits over multi-threaded TRON start to become negligible. Author Feedback Response: While it is fair to assert that the GPU-speedup gains measured for TRON-LR-MIX are consistently higher than the multi-threaded TRON-LR-CPU for all six datasets analyzed in Figure 1, similar assertion cannot be made for TRON-SVM-MIX. The performance of TRON-SVM-CPU (multi-threaded baseline) improves significantly as more threads are added as seen in Figure 2. Measurements on a system with enough parallel threads to push up the performance of TRON-SVM-CPU to its peak would help confirm if it’s actually worse compared to TRON-SVM-MIX. While the data presented in Figure 1 and 2 varies the availability of parallel threads for training, the GPU (NVIDIA Tesla V100) is assumed to be always on at full computational capacity. If the authors could demonstrate speedups for another (less powerful) GPU it would help justify the generalizability of the speedups demonstrated by the paper.

Correctness: As mentioned in the Weaknesses section the claim for tenfold speedup on sparse representation dataset training is not well justified compared to state-of-the-art multi-threaded TRON solver using reasonably large thread count (16-24). Similar issues for dense representation dataset training speedups.

Clarity: The paper does a good job of explaining the nitty-gritties of TRON solver and all the integral compute/data transfer steps it undergoes. Aside from Section 4 and 5 the paper is straightforward and easy to understand. Section 4 and 5 are relatively dense and somewhat difficult to follow. I think referring to the lines of code in Algorithm 1 for specific optimizations described in those sections might help improve the readability of those sections. Table 2 was particularly helpful in understanding the division of steps between CPU/GPU, similar table explaining the breakdown for LR training case for GPU/MIXED optimized TRON solver could have helped improve the clarity of the paper. Also specifying the functionality of the CPU (host) in Table 2 might have been useful. It is understandable that space constraints limited the info provided, maybe this info can be added in supplementary pages.

Relation to Prior Work: The paper clearly explains how past work has mainly focused on optimizing TRON solver using multi-threaded multi-core specific optimizations. The paper presents the first GPU optimized version of the TRON solver which offers significant speedups compared to its multi-threaded past work.

Reproducibility: Yes

Additional Feedback: Line 279-281, disabling parallelization of percolators outer most cross-validation might have worsened the training efficiency of multi-core optimized TRON solver. It would be useful if the authors could share results with parallelization of outer most cross-validation. Author Feedback Response: The authors’ justifications for disabling parallelization of percolator’s outermost cross-validation make sense. This disabling only enabled TRON-SVM-CPU and L2-SVM-MFN solvers to overcome thread scheduling overhead allowing a fair comparison with the proposed TRON-SVM-MIX/GPU. Supplementary material Section 3 describes ‘s’ an asynchronous device stream that plays a key role in concealing large memory transfers between GPU-CPU. I was curious to understand if this stream is utilizing any special changes to generate the results shown in the paper compared to the default setting on the Intel CPU + NVIDIA GPU used for experiments. Typo on line 132, missing bracket for loss function.


Review 3

Summary and Contributions: The authors finished an implementation for TRON (trust region Newton) algorithm on GPUs. The experimental results reported by the authors show that they can achieve 89.2% improvement over single-thread optimized TRON and 65.2% improvement over TRON in the multithread-optimized version of LIBLINEAR for sparse features. For SVM (support vector machines) learning with dense features, they reported they can achieve a 7.6 times speedup, which processed a large dataset with 23 million data instances in 1.9 hours.

Strengths: This is a good engineering paper. The authors provided enough implementation details and the source codes. The structure of this paper is clear. The writing is neat and clean. Overall, this paper is easy to understand.

Weaknesses: In section 4.2, the authors mentioned that z and z^ are completely decoupled of any dependency for host-side computation. In this way, the authors can avoid the large-memory transfer between host and device. Does that means there is a trade-off between memory/computation and communication. If z is huge or extremely dense in some future applications, is this a potential issue? The authors also mentioned decreasing runtimes via mixed-architecture speedups, does that increase the data transfers between host and GPUs? What's the overhead? Will it become an overhead for future applications? The huge speedup of GPU over CPU is not new. In most of the situations, the CPUs are not well optimized [1]. If we want to have a fair comparison between a GPU implementation and a CPU implementation, we may also need to consider the hardware flops peak performance, the hardware price, and the power assumption. A metric like “speedup per dollar” may make more sense in this situation. Also, it is probably not appropriate to just report the speedup given the comparison is based on different platforms. To prove the implementation is great, the authors may need to do an analysis like the roofline model [2]. The authors missed some important references like [3], which was among the first few papers to use GPU to speed up SVM. [1] Lee, Victor W., Changkyu Kim, Jatin Chhugani, Michael Deisher, Daehyun Kim, Anthony D. Nguyen, Nadathur Satish et al. "Debunking the 100X GPU vs. CPU myth: an evaluation of throughput computing on CPU and GPU." In Proceedings of the 37th annual international symposium on Computer architecture, pp. 451-460. 2010. [2] Williams, Samuel, Andrew Waterman, and David Patterson. "Roofline: an insightful visual performance model for multicore architectures." Communications of the ACM 52, no. 4 (2009): 65-76. [3] Catanzaro, Bryan, Narayanan Sundaram, and Kurt Keutzer. "Fast support vector machine training and classification on graphics processors." In Proceedings of the 25th international conference on Machine learning, pp. 104-111. 2008.

Correctness: Yes

Clarity: Yes

Relation to Prior Work: No Some important literatures like [3] are missing. [3] Catanzaro, Bryan, Narayanan Sundaram, and Kurt Keutzer. "Fast support vector machine training and classification on graphics processors." In Proceedings of the 25th international conference on Machine learning, pp. 104-111. 2008.

Reproducibility: Yes

Additional Feedback:


Review 4

Summary and Contributions: The paper deals with the problem of primal optimization. TRON method usually enjoys great speedup, however does not benefit from GPU training due to its sequential dependencies of variable computations. The paper presents a set of methods to address these limits and achieves significant speedup.

Strengths: 1. The paper has great root cause analysis on why TRON does not usually work well with GPU with naive method. The trick proposed by this paper to decouple the variable dependences to avoid large transmission between host and device can be generalizable to other optimizer with similar situations. 2. The empirical results are achieved with testing comprehensive datasets with various settings, which is holistic. 3. It has great speedup on problems like LR and SVM which will be of great interest to the machine learning community who are still using those algorithms as their main classifier.

Weaknesses: 1. The time complexity and memory complexity are not specified, and thus do not give readers crystally clear view on the tradeoffs. 2. Big data mostly lies in industry. While the focus of industry is shifting to deep learning, it is unclear how the proposed method could benefit this large community

Correctness: yes, the claims are correct

Clarity: the paper is well written

Relation to Prior Work: prior works are cited and compared.

Reproducibility: Yes

Additional Feedback: see the strength and weakness sections

[Author Response · NeurIPS 2020]

We thank all the reviewers for their time. In what follows, reviewer comments are italicized and proceeded by our response in blue.

**Reviewer #3**

We thank the reviewer for the helpful references. Importantly, we note that the SVM GPU-speedup paper by Catanzaro et al. is for nonlinear SVMs, which are outside the class of fast, intricate algorithms considered in the paper (see lines 30-51), like TRON.

*Does that mean there is a trade-off between memory/computation and communication. If z is huge or extremely dense in some future applications, is this a potential issue?* The reviewer is correct, there is a trade-off between memory/computation and communication for intricate algorithms like TRON–which contain highly sequentially dependent variables–stated on lines 42-44 and 173-177 (and discussed at length in Sections 4.1-4.2, and lines 240-246). z is a 1D array of length equal to the number of instances, so it will fit in GPU memory even for extremely large problems (as demonstrated in Figure 2). *Probably not appropriate to just report the speedup given the comparison is based on different platforms.* All experiments were conducted on the same platform (stated and detailed on lines 248-249) to ensure a fair comparison. Furthermore, we note that wall clock time (as used in the paper) is the standard metric for speed in machine learning papers, e.g., PyTorch, LIBLINEAR, Cocoa, etc.

*The huge speedup of GPU over CPU is not new.* In general, and in the context of deep learning speedups, we agree. However, GPU speedups for many of the fastest machine learning classification and regression algorithms (such as those in scikit-learn, as well as LIBLINEAR, commonly considered the fastest ML package for such problems) are nonexistent, as discussed on lines 30-39 of the main paper. As stated in Section 1.2.13 of the current scikit-learn documentation:

"Outside of neural networks, GPUs don't play a large role in machine learning today, and much larger gains in speed can often be achieved by a careful choice of algorithms."

The goal of the paper is to show that, contrary to this common conception, GPUs may effectively speedup extremely intricate, fast machine learning algorithms, such as TRON.

**Reviewer #2**

We thank the reviewer for the helpful suggestions on how to improve the readability of Sections 4 and 5, which we will add to the paper.

*The claim for tenfold speedup on sparse representation dataset training is not well justified compared to state-of-the-art multi-threaded TRON solver.* The tenfold speedup is relative to the most widely-adapted, standard LIBLINEAR package (which is optimized for single-threaded use), as detailed in the description of Figure 1. The average reduction in overall runtime compared to the multithreaded CPU optimized TRON is 65.2%, as stated in the bold text on lines 60-61.

*Disabling parallelization of Percolator's outer most cross-validation might have worsened the training efficiency of TRON-SVM-CPU.* The opposite was observed; as noted on lines 268-269, as the number of threads grows large, thread scheduling overhead diminished overall multithreaded performance. With parallelization of Percolator's outer most cross-validation enabled, the overhead of scheduling threads with two nested thread-pools significantly diminished TRON-SVM-CPU and L2-SVM-MFN performance for modest-to-large numbers of threads. In order to fairly measure the performance of these multithread-optimized solvers for large numbers of threads, Percolator's outer-most cross-validation was disabled.

*With reasonably larger number of threads, the speedup claims should be more modest... there exist other platforms with even higher number of threads... It would be interesting to see at what thread count the benefits over multi-threaded TRON start to become negligible.* In practice, solely multithreaded speedups often do not scale linearly in the number of increasing threads, largely due to thread-scheduling overhead (see the above discussion). Thus, even higher thread counts do not necessarily provide more multithreaded performance, as can be seen in Figure 1, where purely multithreaded performance peaks before utilizing the maximum number of threads used in the paper (48) on four of the six datasets. Note that while TRON-LR-MIX similarly displays diminished performance for increased thread counts (as noted on lines 268-720), the relative GPU-speedup gains never become negligible.

**Reviewer #1**

We thank the reviewer for the detailed comments not discussed herein, which we will incorporate into the paper. In addition to lines 30-39, we will include further discussion of previous GPU-offloading work (which, other than non-linear kernels and deep learning, are lacking for fast, intricate classification and regression learning algorithms).

*How generalizable are the proposed optimizations for GPU offloading to algorithms other than LR and SVM?* The general optimization principles described are applicable to any highly specialized, CPU-centric algorithm, e.g., any of the extremely fast algorithms listed on lines 35-36. The specific GPU-optimizations for TRON are generally applicable to arbitrary non-linear losses–such as those widely used in deep learning–through popular automatic differentation packages such as PyTorch and Tensorflow, as stated on lines 150-153 and 313-317.

**Reviewer #4**

*It is unclear how the proposed method could benefit the deep learning community.* As stated on lines 150-153 and 313-317, the detailed TRON GPU-optimizations are readily applicable to arbitrary deep learning losses through popular automatic differentation packages, such as PyTorch and Tensorflow. *The time complexity and memory complexity are not specified.* TRON is a quasi-Newton method, and thus enjoys quadratic convergence while using only linear memory (detailed in the supplementary). Further discussion of complexity and tradeoffs will be included in the paper.

[Meta-Review · NeurIPS 2020]

The paper exhaustively discusses improvements of the TRON algorithm (Trust Region Newton Algorithm), an efficient solver for L2 primal problems, to benefit from both CPUs and GPUs. The improvements are based on increasing parallel processing on CPUs and GPUs, decoupling sequential dependencies of variables, and minimizing the frequency of large memory transfers between CPU and GPU. Overall, this is a solid paper. The reviewers were pointing out that some claims were not justified or misleading, however, the authors succeeded to deliver either an appropriate justification or promised to revise a given claim.